# Recent Progress in the Performance Enhancement of Phase-Sensitive OTDR Vibration Sensing Systems

**DOI:** 10.3390/s19071709

**Published:** 2019-04-10

**Authors:** Romain Zinsou, Xin Liu, Yu Wang, Jianguo Zhang, Yuncai Wang, Baoquan Jin

**Affiliations:** 1Key Laboratory of Advanced Transducers and Intelligent Control Systems (Ministry of Education and Shanxi Province), Taiyuan University of Technology, Taiyuan 030024, China; romzins02@gmail.com (R.Z.); liuxin0127@link.tyut.edu.cn (X.L.); wangyu@tyut.edu.cn (Y.W.); zhangjianguo@tyut.edu.cn (J.Z.); wangyc@tyut.edu.cn (Y.W.); 2College of Physics and Optoelectronics, Taiyuan University of Technology, Taiyuan 030024, China; 3State Key Laboratory of Coal and CBM Co-mining, Jincheng 048000, China

**Keywords:** optical fiber-sensors, Rayleigh backscattering, Φ-OTDR system, vibration measurement

## Abstract

Recently, phase-sensitive Optical Time-Domain Reflectometry (Φ-OTDR)-based vibration sensor systems have gained the interest of many researchers and some efforts have been undertaken to push the performance limitations of Φ-OTDR sensor systems. Thus, progress in different areas of their performance evaluation factors such as improvement of the signal-to-noise ratio (*SNR*), spatial resolution (*SR*) in the sub-meter range, enlargement of the sensing range, increased frequency response bandwidth over the conventional limits, phase signal demodulation and chirped-pulse Φ-OTDR for quantitative measurement have been realized. This paper presents an overview of the recent progress in Φ-OTDR-based vibration sensing systems in the different areas mentioned above.

## 1. Introduction

Owing to their insensitivity to electromagnetic interference, corrosion resistance, electrical insulation, intrinsic safety, good concealment, low cost, etc., optical fiber sensors have demonstrated their superiority over other existing types of sensors [1,2]. Optical fiber sensors, as their name implies, use an optical fiber as the sensing medium. They can be used for static strain, temperature, pressure and dynamic vibration measurements [3,4,5]. Among the various optical fiber sensors, the phase-sensitive Optical Time-Domain Reflectometry (Φ-OTDR)-based sensor system has attracted much more interest for use as a dynamic vibration sensor because of its high sensitivity, large dynamic range, full distribution, simple configuration and relatively easy processing scheme. These types of sensors find applications in the fields of security such as monitoring pipelines, national borders, military bases, state buildings, prisons, embassies, seismic wave detection, etc. Although the Φ-OTDR-based vibration sensor system has demonstrated its supremacy over other types of optical fiber sensors, this has not always been the case. Previously the Φ-OTDR sensor systems were limited by a low signal-to-noise ratio (*SNR*), reduced dynamic range (*DR*), spatial resolution (*SR*) in the range of tens of meters, a tradeoff between the frequency response bandwidth and sensing range, and the nonlinear relationship between the detected vibration and the optical intensity signal, which hindered the use of Φ-OTDR systems for quantitative vibration measurements [6,7,8]. In order to enhance the system performance, many research works have applied different techniques and signal processing methods to push the limitations imposed on Φ-OTDR systems by such factors.

In this paper, we present an overview of the recent progress in the field of Φ-OTDR-based vibration sensor systems in response to the limiting factors mentioned above. The remainder of this paper is structured as follows: firstly we describe the basics operating principle of the Φ-OTDR system. In Section 3, we discuss the methods and techniques for improvement of the *SNR*. Section 4 is dedicated to Φ-OTDR system configurations for very large sensing ranges. Some proposed techniques for improvement of the *SR* and for extension of the frequency response bandwidth of the system are reported in Section 5 and Section 6. The next section addresses the Φ-OTDR system for quantitative measurements, and our conclusions are summarized in the last section.

## 2. Basics Operating Principle of the Φ-OTDR System

The Φ-OTDR system initially proposed by Taylor et al. [9], works by sending highly coherent light pulses into a single-mode fiber and the Rayleigh backscattered light detected by a photo-detector (PD) provides the signal to be processed. A schematic of a Φ-OTDR system is depicted in Figure 1a. A narrow line-width laser is used as a light source to produce continuous highly coherent light destined to be converted into optical pulse signals by an optical modulator (OM) driven by a waveform generator. To prevent the fiber loss, an erbium-doped fiber amplifier (EDFA) can be used to amplify the optical power, then the amplified spontaneous emission (ASE) noise is filtered out before being sent into the fiber under test (FUT) through a circulator. The Rayleigh backscattered light is detected by a PD and processed later by a computer.

Figure 1b is an illustration of the Rayleigh backscattering phenomenon. In fact, the inhomogeneities (illustrated as black spots inside a pulse-width) in the refractive index of the optical fiber due to the imperfect manufacture constitute the scattering centers that act together to produce the Rayleigh backscattered signal detected by the PD. The signal provided by the PD exhibits a speckle-like waveform [10]. Under ideal working conditions of the Φ-OTDR system, it is desirable that the speckle-like trace remains stable until an intrusion to the system induces a noticeable change, so the intensity differential method can be used to determine the intrusion location (see Figure 1c). However, the huge number of the scattering centers inside a pulse-width which confer a stochastic nature to the detected intensity signal, the limiting performance of the laser source, the finite extinction ratio (*ER*) of the OM, the thermal noise in electrical components and sensing fiber, etc., all together influence the accuracy of the sensor performance.

Several parameters are designed to appreciate the performance of a Φ-OTDR sensor system such as the *SNR*, the *SR*, the *DR*, the frequency response bandwidth, the capability for performing quantitative measurements, etc. The first three parameters are extremely related to each other. The *SR* of a measurement is the minimum distance between two intrusions locations allowing them to be clearly non-overlapped. The *SNR* and the *DR* vary in the same direction, as an increase of the *SNR* implies an increase of the *DR*. When considering the discrete model for representation of the Rayleigh backscattering and assuming a rectangular short pulse with a pulse-width of *T*_p_ as input light to the FUT, the detected backscattered signal is formed by the convolution of the probe light with the backscatter impulse response so that the *SR* and the detected optical power *P* can be written as follows [11]:(1)$$\{\begin{array}{c} SR=\frac{v_{gT_{p}}}{2}\\ P=\frac{1}{2}{\alpha v}_{g}{SP}_{0}T_{p}exp\left({-\alpha v}_{g}t\right), \end{array}$$
where *α* is the attenuation coefficient of the fiber, *v*_g_ is the group velocity of the probe pulse, *S* is the backward capture coefficient and *P*_0_ is the input optical power coupled into the FUT. The early methods proposed to improve the *SNR* of the Φ-OTDR system involved increasing the energy coupled into the FUT through the optical pulse by increasing the input optical pulse’s peak power or its pulse-width. Unfortunately, the maximum optical power of the input light is limited by the nonlinear effects described in [12,13]. When reaching the maximum tolerable optical input power, the only one way to increase the *SNR* of the system further is to enlarge the pulse-width of the input signal to the FUT. This operation sacrifices the *SR* of the system. Then, a tradeoff between the *SNR* and the *SR* shows up as a limiting factor for the *DR* of the measurement of the Φ-OTDR system [12].

For a distributed measurement with a Φ-OTDR system and a FUT of length *L*, the pulse period (*PP*) of the input optical power is designed such that its value is greater than or equal to the round-trip time of the light in order to prevent the pulse overlapping: (2)$$PP\geq \frac{2L}{v_{g}}$$

In a Φ-OTDR system, every pulse propagating inside the FUT corresponds to one sampling of the external vibration and the pulse repetition rate should determine the vibration sampling rate. This condition combined with the Nyquist’s sampling condition imposes the maximum measurable vibration frequency to be:(3)$$f_{\max}=\frac{1}{2PP}\leq \frac{1}{2}\left(\frac{v_{g}}{2L}\right),$$
while the minimum detectable vibration frequency is determined by the noise present in the system.

## 3. Methods and Techniques for Improvement of the *SNR* in Φ-OTDR Sensor Systems

For a given *SR* (probe pulse-width), several Φ-OTDR-based vibration sensor schemes have been proposed to increase the *SNR* and the *DR* of the vibration measurement not necessarily by increasing the input probe light peak power but by reducing or controlling the noise present in the system [14,15,16,17,18,19,20,21,22,23,24,25,26,27,28,29]. In [14], a coherent detection-based Φ-OTDR system has been proved to have a better *SNR* than that of a direct detection system. In the coherent detection, the laser output light is first split into two beams by an optical coupler and later another optical coupler is used to recombine the backscattered signal with the reference light. With the coherent detection scheme, there might be polarization mismatch between the Rayleigh backscattered signal and the reference light signal, which can affect the vibration measurement, so the coherent detection was associated with the moving averaging and moving differential technique to de-noise the detected signal. The moving averaging and moving differential technique consists of acquiring a certain number N of the Rayleigh backscattering traces and choosing M number of the acquired traces to be averaged. Thus, N-M+1 subsets of averaged traces can be obtained with the noise power in the measurement reduced by a factor of 1/M. These averaged traces with reduced noise power constitute later the signals to be analyzed. Experimental results demonstrate that for a given optical input power, the *SNR* increases with the averaging number. Juarez et al. proved experimentally the influence of the laser frequency drift on the detected signal and concluded that the lower the frequency drift of the laser is, the better the system performs [15]. The laser frequency drift even lower, impacts on the received Φ-OTDR trace in the sense that it will show some trace-to-trace fluctuations [16]. These trace to trace fluctuation act as a noise that fixes the minimum detectable vibration frequency to a higher value. The authors in [17] proposed the wavelength diversity technique to actively compensate the influence of laser source frequency drift, which enabled them to detect a vibration of 0.5 Hz frequency under a laser source frequency drift of mean value of 1.68 MHz/s. The basis of the method is the use of a wavelength tunable laser source to modulate the laser output frequency through the parameters sweep number *N* and frequency sweep step *Δf*. Figure 2 describes the working principle of the method: at an instant of time *t*_0_, the laser output frequency is *f*_0_ and an initial Φ-OTDR trace is recorded; at any other instant of time, because of the laser frequency drift the laser output frequency is *f* different from *f*_0_. When sweeping the laser output frequency from *f − NΔf* to *f + NΔf*, 2*N* + 1 traces are recorded and the one best correlated to the previous trace is found out in such a way that the two highly correlated traces can be considered as arising from the same laser frequency output. Doing this continuously enabled them to suppress the influence of the laser frequency drift and make it possible to locate a vibration with frequency as low as 0.5 Hz. The veritable problem with this method is the relatively long time needed for data transmission and processing which reduced significantly the dynamic performance of the system. 

Another important source of noise in Φ-OTDR systems, which needs to be controlled, is the finite *ER* of the OM. During the process of conversion of the continuous light from the laser into optical pulses, there is always a part of the continuous leakage light that cannot be suppressed and the received signal is made up of the pulse and the continuous backscattered light parts, respectively. The leakage light part of the Rayleigh backscattering signal acts as a noise that affects the sensitivity of the Φ-OTDR system [18,19,20]. The *SNR* of the system is linked to the *ER* of the optical modulator as [18]: (4)$$SNR\left(z\right)=\sqrt{ER}\sqrt{\frac{sinh\left(\frac{v_{g}T_{p}\alpha }{2}\right)exp(-2\alpha z)}{2\left(1-exp(-2\alpha L)\right)}}$$
where *z* denotes the position in the FUT, the others parameters being defined in the previous section. The expression (4) shows theoretically that for a given *SR* (constant optical pulse-width *T*_p_), the *SNR* and the *ER* vary in the same direction. For the purpose of corroborating the theoretical demonstration with experimental results, some authors proposed different methods for generation of higher *ER* pulses and implemented them in a Φ-OTDR system for comparison with the conventional configuration (see Figure 3). The methods for generation of the higher *ER* pulses are respectively the nonlinear optical loop mirror (NOLM), the association of two electro-optic modulator (EOM) [18], and the nonlinear Kerr effects on the sinusoidally modulated optical signals (SMOS) methods [20]. The experimental results indicate in the case (b) that, when the pulse is off and only the continuous leakage light is coupled into the FUT, the noise floor in the system with NOLM (*ER* = 60 dB) is 11 dB lower than the system with conventional pulse modulation scheme one (*ER* = 30 dB). In (d), the principle of generation of high *ER* optical pulses is based on the fact that a SMOS passing through a nonlinear Kerr medium experiences the self-phase modulation and generates at its output some new spectral components such that any small change in *ER* of the pulses at the input of the Kerr medium can cause large variation in the *ER* of the different components at the output of the Kerr medium expressed as [20]:(5)$$\varepsilon_{\mathrm{out}}^{\left(n\right)}\left[dB\right]=\left(2n+1\right)\varepsilon_{\mathrm{in}}\left[dB\right]-3.2n-2.7;\;5\;dB<\varepsilon_{\mathrm{in}}<40\;dB$$
where $\varepsilon_{\mathrm{in}}$ is the *ER* of the pulse signal input to Kerr medium, $\varepsilon_{\mathrm{out}}^{\left(n\right)}$ is the *ER* of the n^th^ order sideband component of the output, *n* being an integer ranging from 1 to 6. In (d), the two cascaded EOMs are driven respectively by sinusoidal and square pulse electrical signals to form an impressed square-pulse SMOS, then the Kerr medium causing the self-phase modulation phenomenon to produce the sidebands that are sent later to a BP optical filter to select the desired component with higher *ER*. The method has been regularly implemented in Φ-OTDR systems and its influence on the system performance has been effectively demonstrated [20]. As for the work [19], the relationship between the *ER* of the OM and the *SNR* has been investigated using the effect of the variation of the diffraction efficiency. The *ER* of the OM has been changed many times by varying the driving voltage of the OM, and the optical intensity change caused by the disturbance has been recorded. Experimental results indicate the intrusion miss in the cases of *ER* ≤ 8.7 dB even when combined with the moving averaging and differential averaging to de-noise while, the *SNR* of the system scales directly with the *ER* of the OM. 

In [21], the authors instead analyzed the influence of the pulse shape on the *SNR* of the detected signal and found out that the adopted rectangular-like pulses are more conductive to nonlinear effects and Gaussian or triangular shaped pulses perform better and offer a *SNR* nearly the double of that provided by the rectangular pulses under the same working conditions. In addition to the noise in the system caused by the limited performance of the devices such as laser, optical modulator, optical receiver, there is also a noise arising from the behavior of the polarization of the light propagating along the FUT. This noise source is responsible for the polarization-induced fading, a common limitation in any fiber-optic interferometric system. In a Φ-OTDR system, the state of polarization (SoP) of the Rayleigh backscattered light from a scattering center to another within a pulse-width of the probe pulsed light cannot be maintained constant. This fluctuation of the SoP of the backscattered light affects the visibility of the detected signal. The polarization induced fading demonstrated in Φ-OTDR system in the case of direct detection [22], is much more serious in the coherent detection-based system since there is polarization mismatch between the Rayleigh backscattered signal and the reference light signal. Consequently, the polarization maintaining fibers configurations systems shouldn’t be subject to that phenomenon [23]. In [24], the authors adopted the polarization diversity technique to overcome the polarization induced fading problem in a coherent Φ-OTDR system (Figure 4). For that purpose, a polarization beam splitter is used to divide the detected signal into orthogonal polarization states components and processed differently. The experimental results demonstrate for each of the processed signal an improvement of the *SNR* of respectively 10.9 dB and 8.95 dB.

Recently, Muanenda et al. proposed the introduction of the cyclic pulse coding in Φ-OTDR systems, which enables one to send a higher quantity of energy in the system while avoiding the nonlinear effects so that the *SNR* of the system can be improved [25]. The basic idea subtending the pulse coding is that measuring several objects together instead of separately may make it possible to determine them more accurately than the individual measurements. The key of the coding technique in [25] is the S-code. The procedure consists of two steps: the pulse encoding and pulse decoding. During the pulse encoding, the authors injected into the FUT a sequence S-coded rectangular pulses while avoiding interference from two consecutives pulses such that the resulting backscattering for any position in the FUT is the integration from that of many pulses. The pulse-decoding step is to obtain the equivalent single-pulse backscattering trace by multiplication of *S*^−1^ and the vector of the backscattering trace from many pulses, so the improvement in *SNR* compared to the single-pulse can be evaluated. The implementation of cyclic pulse coding in a Φ-OTDR system (Figure 5) requires the laser light to be coherent within a pulse-width and incoherent outside to assure the linearity of the optical intensities addition. These conditions associated together with the expression of the laser coherence length for a Lorentzian shape imposed on laser light line-width to satisfy:(6)$$\frac{N_{\max}}{{\pi \times R}_{\mathrm{TT}}}\leq \Delta \vartheta \leq \frac{1}{{\pi \times T}_{p}},$$
where *N*_max_ the S-code length, *R*_TT_ the round-trip time of the laser light in the FUT, *T*_p_ is the pulse-width, Δϑ being the laser line-width. This restriction on the laser line-width made the low-cost DFB laser a good candidate for the experiment of the pulse coding in Φ-OTDR system. In the experiment, a pair (MZM, AOM) assured the production of cyclic pulses, first the MZM is used to produce continuous pulses in ‘1’ state and later the AOM to generate a high *ER* light signal at his output. The advantages of the proposed system include low-cost compared to the conventional Φ-OTDR using a highly coherent laser source and offering, an improvement in the *SNR* of nearly 9 dB without affecting the frequency response bandwidth of the measurement.

Shao et al. suggested an adaptive 2D bilateral filtering algorithm in a Φ-OTDR system to enhance the *SNR* [26]. The proposed bilateral filtering algorithm uses the weighted average with the geometric weight reduced to a simple Lorentz function so that tuning the gray level standard deviation alone can be sufficient to remove the noise in the system without altering the desired signal. The proposed method is implemented in coherent detection scheme, an improvement of ~14 dB of the *SNR* with a sensing range of 27.6 km has been experimentally demonstrated without deterioration of the *SR*. For faster signal processing, the filtering window size was set to 20 and the double iteration was preferred since there is no consequent *SNR* improvement for greater filtering window size, while higher filtering window size consumes the processing time. Better *SNR* improvement with the default of relatively long signal processing time has been proposed before based on empirical mode decomposition (EMD) method in [27]. In this method, the raw Rayleigh backscattered signals were first decomposed into intrinsic mode functions (IMFs) and a residual component through the sifting algorithm. Later, the Pearson correlation coefficient between the signal and the IMF is found out so that the high frequency noise component will be progressively eliminated through filtering. The method demonstrated a *SNR* of more than 40 dB in a sensing range of 2km to detect respectively 100 Hz and 1.2 kHz vibrations. Many others signal processing-based methods to enhance the *SNR* in Φ-OTDR system have been also proposed such as wavelet de-noising method [28], 2D edge detection [29], etc.

## 4. Configurations of Φ-OTDR Systems for Large Sensing Range

An important parameter limiting previously the practicability of the Φ-OTDR system is its reduced *DR*. For a given *SR*, increasing the optical peak power input to the FUT via an EDFA causes an increase of the *DR* and the *SNR* of the system while the input optical peak power cannot be indefinitely increased due to the nonlinear effects. Consequently, this method cannot improve significantly the *DR* of the system. Most of the Φ-OTDR sensor systems rely on the optical amplification for consequent extension of the *DR*.

Rao et al. proposed the use of the combined action of an EDFA and the Raman optical amplification technology to extend the sensing range in a Φ-OTDR system to 62 km with a *SR* = 100 m [30]. The proposed experimental setup is shown in Figure 6. For their experiment, the authors adopted the bi-directional Raman amplification with a gain equalization offering even distribution along the sensing fiber than any other scheme. The bi-directional Raman pump power evolution is governed by the following equations [31]: (7)$$\{\begin{array}{c} \frac{{dP}_{P}}{dz}=g_{R}P_{R}P_{P}-\alpha_{P}P_{P}\\ \gamma \frac{{dP}_{R}}{dz}=-\frac{\omega_{R}}{\omega_{S}}g_{R}P_{R}P_{P}-\alpha_{R} \end{array},$$
where *P*_p_, *P*_R_ are respectively the pulse and pump power; $\alpha_{R}$, $\alpha_{P}$ are respectively the transmission losses of the pump and the signal, $\omega_{R}$, $\omega_{S}$ are respectively the frequencies of the pump and signal light respectively, $g_{R}$ is the Raman gain and *γ* = +1(−1) depending on the pumping direction. 

Later, the authors implemented the method in a coherent detection-based Φ-OTDR system. In this scheme, the heterodyne detection removes the ASE noise caused by the high power of the Raman pump and a wise choice of the probe pulse power and the Raman pump power enables them to avoid the SBS and realize a 131.5 km long sensing range and 8 m of *SR* in the Φ-OTDR system [32]. Furthermore, the same research group utilized a hybrid amplification scheme made simultaneously of co-pumping second-order Raman amplification, counter-pumping first-order Raman amplification and counter-pumping Brillouin amplification with the caution that each pumping scheme controls a certain region of the FUT [33]. Table 1 lists the working conditions of the different pumping schemes. With this method the sensing distance is enlarged to 175 km with a *SR* of 25 m and a *SNR* = 12 dB, the experiment was carried out with a highly coherent laser whose line-width is 100 Hz. The laser output light is split twice by two optical couplers to create the reference light and two inputs probe light for both ends of the sensing fiber, a microwave generator driving an EOM maintains the frequency gap between the input light and the Brillouin pump while light depolarization helped to suppress the polarization gain dependent influence. Experimental results demonstrate the necessity of complicity between the different pumping schemes to keep the signal power high enough along the whole sensing distance.

Martins has reported comparative studies of the use of the first and second-order Raman amplification to pump the Φ-OTDR system and demonstrated experimentally that the second-order Raman amplification offered clearly better *SNR* to the system [34]. The optical amplification used as pump for extending sensing range should reduce the degree of freedom of the system and in this sense a Φ-OTDR sensor system in its original structure with relatively long sensing distance will be more desirable. Pang et al. used the standard deviation of the amplitude of the demodulated signal in a coherent detection based Φ-OTDR system with high speed data acquisition card (12-bit, 1 GS/s) to detect the intrusion in a 50 km long FUT with a *SNR* of nearly 32 dB [35].

## 5. Φ-OTDR System with Spatial Resolution in the Sub-Meter Range

In conventional single pulse Φ-OTDR systems, the *SR* is determined by the half of the pulse-width of the probe light so the narrower the pulse is, the better the resolution of the system is, while in the same time reducing the pulse width of the probe light implies lower energy sent into the FUT leading to the degradation of the sensitivity of the measurement. A common approach to this limitation is the pulse coding method (described in Section 3) that improves the *SNR* through the energy sent into the FUT while keeping the *SR* unchanged. Recently, some techniques involving mainly the linear frequency modulation (LFM) pulse-compression and the chirp pulse amplification (CPA) emerged to break totally this dilemma in the Φ-OTDR system and enabled to improve significantly the *SR* [36,37,38,39,40]. The LFM pulse-compression inspired from the radar communication, is utilized here in a coherent detection based Φ-OTDR system. The signal detected by a BPD is I/Q demodulated to obtain complex analytic signal from real valued detected signal which will pass later through a matched filter to produce the OTDR trace [36,37,38]. Theoretical demonstrations show that the pulse-compression modifies the LFM pulse signal into a sinc-like profile with its main lobe strongly related to the sweeping range that determines the *SR* of the system. Figure 7 is the setup of the LFM pulse-compression technique in Φ-OTDR system [36] (also called optical pulse-compression reflectometry (OPCR)) and the module (a) was used to generate the LFM pulsed in a first experiment. The pulse-compression operation was conducted as follows: a single sideband modulator (SSBM) linked to a voltage-controlled oscillator (VCO) with 221 MHz sweeping range driven by sawtooth waveform modulates the laser light output into LFM signal which is sent later to a MZM driven by a square pulse synchronized with the VCO to produce the pulsed LFM signal. The authors demonstrated experimentally a *SR* of 47 cm in a sensing range of 5.4 km with the method. Later, a system with increased stability has been proposed by the same research to group to improve the *SR* to 10 cm with the same sensing range [37]. The new method for the generation of the pulsed LFM signal (see module (b)) suppresses the need of a MZM and used only the VCO under the simultaneous action of a sawtooth and a rectangular signal originating from an AWG to drive the SSBM. In this case, the pulsed LFM has better *ER* that impacts on the system performance as described in Section 3. In [38], Cai et al. proposed a Φ-OTDR system with frequency swept-pulse based on the same principle as described previously. Here, the LFM signal after production is sent into a DFB-LD by injection locking to enhance the *ER* and the sweeping range of the LFM signal which is transferred later to an AOM synchronized with the VCO to generate the pulsed LFM signal. Once generated, the pulsed LFM signal is amplified and launched into the FUT. With this scheme the LFM pulse frequency bandwidth was extended to 420 MHz and a Φ-OTDR system with a *SR* of 30 cm and a sensing range as long as 19.8 km has been realized for the first time.

Basing their analysis on the fact that the noise sources such as the random nature of laser phase, the light SoP mismatch, the I/Q quadrature imbalances may affect the numerical compression in the OPCR, Pastor-Graells et al. proposed a Φ-OTDR system using the chirp pulse amplification to realize the optical domain pulse-compression [39]. The proposed chirp pulse amplification Φ-OTDR system is the conventional Φ-OTDR system including two linearly chirped fiber Bragg gratings (LC-FBGs) with opposite dispersion coefficient. The first LC-FBG stretches an ultra-pulse in the range of picosecond which is launched into the FUT after amplification and the Rayleigh backscattered signal is compressed with the second LC-FBG (see Figure 8). In fact, in the conventional Φ-OTDR system, the backscattered signal trace is expressed in both the time and frequency domains as [39]:(8)$$\{\begin{array}{c} I\left(t\right)\sim p\left(t\right)*f\left(t\right)\\ I\left(w\right)=P\left(w\right)\times F\left(w\right), \end{array}$$
with *I*(*w*), *P*(*w*) and *F*(*w*) being the Fourier transform of respectively *I*(*t*), *p*(*t*), and *f*(*t*) defined previously in Section 2. After the combined action of the pulse stretching and compression of the detected signal trace then we have respectively in the frequency and time domains:(9)$$\left\{ \begin{array}{c} I^{\prime }(w)=G\times P(w)\times F(w)\times exp\underset{\mathrm{pulse}\;\mathrm{stretching}}{\underset{︸}{\left[+j(\frac{\ddot{\Phi }}{2})w^{2}\right]}}\times \underset{\mathrm{trace}\;\mathrm{compression}}{\underset{︸}{exp[-j(\frac{\ddot{\Phi }}{2})w^{2}]}},\\ I^{\prime }(t)=G\times I(t), \end{array}\right.$$
where *I′* is the signal intensity in the proposed system, $\ddot{\Phi }$ is the second-order dispersive coefficient of the LC-FBG and *G* the amplification gain factor. The *SNR* of the backscattering trace is improved according to the amplification gain factor *G* while the *SR* is kept same as before pulse stretching. A *SR* of 1.8 cm in a sensing range of 8 m limited by the high repetition rate of the locked laser to generate the coherent narrow pulses with an increase of ~20 dB in the *SNR* compared to the conventional Φ-OTDR system. The demonstrated temporal width was wider than the pulse-width of the probe light because the 35 GHz bandwidth of the PD used in the experience is lower than the pulse bandwidth. Later, the authors improved the *SR* of the system to 3 mm by replacing the PD with that of 45 GHz bandwidth [40].

## 6. Φ-OTDR Systems with Frequency Response Bandwidth Higher than the Limit Set by the Sensing Range

In this section, we discuss the proposed schemes for improvement of the frequency response bandwidth in Φ-OTDR systems. As mentioned in the Equation (3), the extension of the sensing range of the system implies a higher *PP* of the probe light and a reduced frequency response bandwidth of the system. There is then a manifestation of a tradeoff between the frequency response bandwidth and the sensing range of the system while some industrial demands need a Φ-OTDR system with long sensing range capable of monitoring high-frequency vibration. To overcome this limitation Cai et al. introduced the frequency division multiplexing (FDM) technique in Φ-OTDR systems to build the so called FDM Φ-OTDR [41]. Previously the FDM has demonstrated its capability to improve the sensitivity of a COTDR system [42,43]. In the FDM Φ-OTDR system concept, a probe pulse sequence made of *n* evenly spaced frequency components is launched into the FUT in such a way the frequency step is high enough to avoid the signal interactions. Then, the *n* Rayleigh backscattered traces corresponding to each frequency component can be detected proving that the signal detection rate is *n* times greater than that of the conventional Φ-OTDR system. Later, the signal demodulation is processed to obtain the phase information of the corresponding detected Rayleigh backscattered light. The variance and the spectrum of the phase extracted signal will give respectively the location and the frequency of the external vibration applied to the PZT. For their experiment, the technique was implemented in a coherent detection-based system. To produce the probe pulse sequence light, an EOM driven by an electrical signal with its frequency changing in a staircase pattern generates at its output a multi-frequency signal which is in turn pulse-modulated by an AOM. The effectiveness of the FDM Φ-OTDR was firstly demonstrated to detect a 3.4 kHz square wave located at 9.5 km of the FUT with four frequency components (*n* = 4). The multiple backscattered signals corresponding to different frequency components is acquired with a DAQ at a sampling rate of 500 MHz/s, then the phase demodulation is performed to restore the external applied vibration. The FFT results of the reconstructed vibration confirmed the presence of the applied vibration frequency and its harmonics up to 20 kHz, what is not detectable with the conventional single-frequency pulse Φ-OTDR since the repetition rate limit should be 5 kHz in a FUT of 10 km length. With this method, the number *n* of frequency components in the probe pulse sequence can be changed depending on the DAQ rate and the capacity of the computer used for data processing.

A FDM Φ-OTDR system performing better than the one described previously has been suggested by Iida et al. [44]. For their experiment, the authors used a 1.25 GHz/s DAQ as the signal receiver and the pulse sequence was made of *n*=16 frequency components with the caution to prevent the SBS. To measure the vibration, the short-time Fourier transform is applied to the multiple back-scattered detected signals to obtain *n* OTDR traces. After the propagation of a certain number *k* of pulse sequences into the FUT, the n×k OTDR traces denoted as *A_ij_*(*t*) (1≤i≤n, and 1≤j≤k) are arranged chronologically in the time-domain and a FFT is applied to the data for the different positions to extract the vibration information. Figure 9 is an illustration of the signal processing procedure. In this experiment, the authors successfully extracted an 80 kHz vibration along a FUT of 5 km, which is over the limit set by the repetition rate in a conventional Φ-OTDR.

More recently, the FDM method has also been implemented in a direct-detection Φ-OTDR system by Fan et al. to address the dilemma between the frequency response bandwidth and the measurement range [45]. The method works by using a frequency step sweeping laser source and modulating the AOM such a way to generate a dual-pulse pairs with different frequency-shifts. A low-bandwidth avalanche photo-detector (APD) should eliminate the interference signal between the dual-pulse pairs and the use of the phase extraction method will allow to recover the external vibration. Figure 10 shows the illustration of the setup for the FDM dual-pulse Φ-OTDR. In their experiment, a NLL combined with an intensity modulator designed to introduce an appropriate frequency offset acts as a tunable laser source and the AOM to generate a sequence of four raised-cosine shaped pulse-pairs with different frequency spacing to probe the FUT. It’s worth to mention that the pulse shape here is changed from the conventional rectangular-shaped pulse in order to remove the influence of the crosstalk due to the double-frequency probe signal on the detection. The detected signal by the low-bandwidth APD has been passed through four digital BPFs whose center frequencies correspond respectively to the spacing frequency and the beat signal corresponding to the four intensity signals are used for the phase extraction. The method demonstrates with success the recovery of an external vibration with its frequency ranging from 1 to 17 kHz with a flat response for a sensing range of 10 km. It should be noted that the maximum detectable frequency for the conventional direct-detection Φ-OTDR system for such sensing range should be 5 kHz according to Equation (3). 

## 7. Φ-OTDR Systems for Recovery of the Full Vector Information of External Vibrations

After being able to detect the presence of the intruder, its location and frequency components, a Φ-OTDR system capable of recovering the amplitude and the waveform of the vibration applied at any point of the FUT is of great interest since it can find more applications in the security areas. The differential intensity of the Rayleigh backscattered signal does not have a linear relationship with the amplitude of the external disturbance while the phase change induced by the disturbance through the relation *Δφ = Δn·k·L* does respond linearly to the refractive index change induced by the disturbance. The quantities *n*, *k* and *L* represent the refractive index of the optical fiber, the wave vector amplitude of the laser light and the length of the FUT respectively. Recently, a Φ-OTDR system for obtaining the full vector information of the external vibration has been proposed [46,47,48,49,50,51,52,53,54,55,56,57,58,59,60,61,62,63]. In [46], the authors employed the digital coherent detection method in a Φ-OTDR system to obtain the amplitude and the phase of the detected signal and the difference of such information in two points (before and after the disturbance region) allowed them to detect the location and the waveform of a 200 Hz vibration frequency with a *SR* of 5 m over a FUT of length 3.5 km. However, this method requires the analysis of a large amount of data and the demodulation result was also very sensitive to the coherent Rayleigh noise. To eliminate the Rayleigh fading, Masoudi et al. introduced a MZI associated with a 3 × 3 symmetric coupler at the output at the receiving end of the Φ-OTDR system and the polarization diversity technique has been used to recover the phase change induced by the disturbance [47]. The concept of the phase shift induced in the detected optical intensity signal is illustrated in Figure 11. The technique has demonstrated experimentally a phase demodulation with a frequency response bandwidth of [500–5000 Hz] and respectively a *SR* of 2 m and strain resolution of 80nε. The main problem here is that the system’s stability is determined by the MZI which is very sensitive to the ambient temperature variation and also, there is a need of three sensitive PDs working synchronously at the receiving end of the system.

Later, Alekseev et al. proposed a scheme maintaining the simple configuration of the Φ-OTDR system using the dual pulsed phase modulated light to probe the FUT and the phase diversity technique to rebuild the applied vibration [48]. The principle of the technique is to generate cyclically three groups of pulse pairs in such a way the second pulse of each sequence is respectively 0, +2π/3, −2π/3 phase shifted by a phase modulator (see Figure 12), so that three groups of Rayleigh backscattered signals phase shifted in the same order with the pulse pair sequences are detected and processed according to the phase diversity technique. The technique was consistent to recover different kinds of vibration. The drawback of this technique is that the system’s frequency response bandwidth is reduced by 1/3. In [49], Wang et al. thought of the I/Q demodulation technique in a coherent based Φ-OTDR to reconstruct the external vibration applied to the FUT; an acoustic frequency-shifter is placed in the local light branch to compensate the phase shift introduced in the signal branch by an AOM, so that the system is a homodyne detection. Then, the local light and the Rayleigh backscattered signal are input to a 90° optical hybrid which generates at its output two quadrature components *I*(*t*) and *Q*(*t*). The I/Q components are processed to recover the induced phase shift *φ*(*t*) according to the I/Q demodulation algorithm. Figure 13 is the experimental setup of the system.

Dong et al. also used the I/Q demodulation technique in the coherent detection system [50]. The system here is the heterodyne detection. The two-quadrature components *I*(*t*) and *Q*(*t*) are related to the phase of the backscattered signal and the strain by:(10)$$\{\begin{array}{c} \varphi \left(t\right)=\arctan\left(\frac{I\left(t\right)}{Q\left(t\right)}\right)+m\pi ,\\ \frac{\Delta \varphi_{\mathrm{ij}}}{z_{\mathrm{ij}}\Delta \varepsilon }=\frac{4\pi }{\lambda }\left({n+C}_{\varepsilon }\right), \end{array}$$
Δε is the precise amount of the strain applied to the fiber with a nanopositioning translation stage along a distance *z*_ij_, *n* and *C_ε_* being respectively the refractive index of the fiber and its strain parameter, *λ* is the laser output wavelength, and *m* represents any integer. The conditions of the experiments were very strict (at night on the first floor to avoid any environmental disturbance) and close attention was paid to the polarization-induced noise through the use of the polarization-maintaining fiber and polarization beam splitter. With this method the authors demonstrated experimentally a precise dynamic strain measurement in the range 10–1000nε with a strain resolution of 1 or 2nε according to the *SR* of 2.5 m or 5 m.

Early methods used for precise quantitative measurement of temperature/strain in Φ-OTDR systems were based on the compensation of the refractive index change induced *Δn* by a frequency shift *Δυ* of the pulse launched into the FUT. This method progressed from static temperature/strain measurement to dynamic measurement. In work [51], Koyamada et al. considered the Rayleigh backscattered light as the interference of two signals such that one is sensitive to the temperature/strain while the second corresponds to the distribution of the scattering centers in the unperturbed FUT. To measure the temperature/strain, the FUT is divided into many sections subjected to two different controlled temperatures *T*_a_ and *T*_b_ and the Rayleigh backscattered signals *P*_a_(*υ, z*) and *P*_b_(*υ, z*) were performed and compared through their cross-correlation value. When the two values of temperature are equivalent, the signals *P*_a_(*υ, z*) and *P*_b_(*υ, z*) are highly correlated, by contrast when *T*_a_ is different from *T*_b_ then *P*_a_(*υ, z*) is highly correlated to *P*_b_(*υ+Δυ, z*), where *Δυ* is the frequency shift that compensates the variation in temperature and strain as expressed in the relations (11):(11)$$\{\begin{array}{c} \frac{\Delta \upsilon }{\upsilon_{0}}=-0.78\times \Delta \varepsilon \\ \frac{\Delta \upsilon }{\upsilon_{0}}=-\left(6.92\times 10^{-6}\right)\times \Delta T \end{array},$$
$\upsilon_{0}$ is the central laser frequency while *Δε* and *ΔT* represent the strain and temperature change respectively. The technique demonstrated a measurement with a temperature resolution of 0.01° and a *SR* of 1 m over a sensing range of 8 km. A SSBM driven by a synthesizer was used to apply a precise frequency shift to the probe light. The operation of the frequency shift compensation and the averaging of numerous traces in order to de-noise the signals needed very long time (some hours), which rendered the measurement static. Zhou et al. also developed the same concept as in [51]. To compensate the induced frequency shift, the authors used an EOM driven by a microwave source associated with a tunable filter to filter out one of the sidebands of the modulated light in a direct detection scheme which is less subject to polarization induced noise than the coherent detection [52]. Doing this reduced the frequency sweep time needed for compensation of the induced refractive index change to 0.04s, so they demonstrated this time a quasi-static 10nε strain resolution measurement. Recently Pastor-Graells et al. showed an improvement in the technique by using a linear chirped-pulse instead of the single-frequency pulse in the conventional Φ-OTDR system in the direct detection scheme [53]. During the pulse propagation, the frequencies components of the pulse vary with the position in the FUT, so that any refractive index change *Δn* induced by any disturbance will shift longitudinally the same Φ-OTDR trace pattern by *Δz* such that the frequency shift *Δυ* compensator of the change *Δn* (respectively the temperature/strain) can be deducted from *Δz*. This method avoids the time-consuming frequency sweep operation and then impacts on the dynamic nature of the measurement. The longitudinal shift *Δz* is found out through the cross-correlation of two consecutive detected signals and continuously to enlarge the temperature/strain measurement range. The linear chirped-pulse has been experimentally realized by driving the laser source with a repetitive ramp electric signal. With this technique, the measurement is relative and the measurement range can be extended progressively, experimental demonstrations showed 1 mK/4nε resolution with a *SNR* > 25 dB. The limitation with this method is that the detectable change *Δn* from trace to trace should be in a certain precise range to avoid cumulative errors and the high digitization speed needed to cover the total bandwidth of the chirp signal used. Later works of the same research group investigate the impact of the laser line-width on the accuracy of the temperature/strain measurement [54,55]. As mentioned in the Section 3 of the manuscript, the laser source frequency drift induces random temporal shift from trace to trace even in the unperturbed region of the FUT and the extent of the fluctuation depends on the degree of coherence of the laser. This random temporal shift will determine the accuracy of temperature/strain measurement when using the chirped-pulse Φ-OTDR. This fact has been corroborated experimentally by changing the laser source with different line-widths. Experimental results demonstrate an improvement of *SNR* of more than 22 dB by changing the laser line-width from 5 MHz to 25 kHz [54]. Removing the influence of the finite laser line-width on the stability of the unperturbed region of the FUT from the measured quantities in the perturbed region of the FUT allows more accurate measurement [55]. Another work studies the extension of the sensing range with the assistance of the first order bi-directional Raman amplification [56]. As the important thermal noise in the high-bandwidth PD used in the chirped-pulse Φ-OTDR system can facilitate the relative intensity noise transfer from the Raman pump to the detected signal associated with the necessity to avoid the nonlinear effects of the modulation instability, there is need to pay careful attention to the optical pulse peak-power and the Raman pump powers. Taking these two points into account, the injected pulse peak-power has been chosen to be 25 mW and the co-propagated and the contra-propagated Raman pumps are 350 mW and 250 mW. The achieved performance in those experimental conditions is: the sensing range enlarged to 75 km, strain recovery with a *SNR* > 20 dB while, the conventional chirped-pulse Φ-OTDR limit is 17 km. The linear chirped pulses Φ-OTDR system brings some new features that deserve close attention for studies. 

By contrast, Feng et al. suggested the phase demodulation method in Φ-OTDR without coherent detection utilizing both single and dual-pulse probe schemes [57]. To do so, the authors modeled the cosine-relationship of the backscattered signal in the disturbance region as:(12)$$I_{i}\left(t\right)=D_{i}+A_{i}\cdot \cos\left(\theta \left(t\right)+\Psi_{i}\right),$$
where *D_i_*, *A_i_* and $\Psi_{i}$ are slowly varying quantities, *θ*(*t*) represents the phase to be recovered. The phase demodulation here is conducted as follows: firstly, normalize the detected intensity at two adjacent points in the disturbance region, process the sum and the difference of the normalized version of these intensities and then a second normalization to generate the I/Q quadrature components. Finally, the phase signal *θ*(*t*) can be recovered as requires the I/Q demodulation. The authors have successfully tested this method with three different kinds of vibrations: single-frequency vibration, amplitude-modulated vibration and chirped vibration. The main problem with this method is the influence of the coherent Rayleigh fading which generates some undesirable spikes in the demodulated phase signal.

In [58], the authors proposed the phase generated carrier (PGC) demodulation technique in the Φ-OTDR system, a PZT driven by an AWG in one arm of an unbalanced MI introduced at the receiving end of the system is used to generate the carrier signal and the Faraday rotator mirrors at the MI end played the double role of light reflection and elimination of the birefringence effects of the fiber on the SoP of the light input to the MI (see Figure 14). The detected light is an interference of the lights recombined by the optical coupler with a time delay since there is a difference in the optical path length and expressed as: (13)$$I_{i}\left(t\right)=I_{D}+I_{C}\cdot \cos\left(C\cdot \cos(wc\cdot t+\theta (t)\right),$$
where *I_D_* and *I_C_* are respectively the dc current and the amplitude of the ac current, *C* and *wc* are respectively the carrier modulation depth and frequency, θ(t) the phase to be recovered and later the phase demodulation is performed according to the PGC-Atan algorithm. The experiment demonstrated a system with a *SR* of 6 m, a sensing range of 10 km and a *SNR* of 30.45 dB. This method is insensitive to light intensity disturbance since the demodulated result is independent of *I_D_* and *I_C_*. Two limitations can be noted for this approach: (1) the MI at the receiving end of the system is temperature sensitive, and (2) the implemented demodulation technique imposes on the carrier modulation depth the value *C* = 2.63 so that J_1_(*C*) = J_2_(*C*) while keeping the modulation depth strictly constant is still a challenge. In [59], the authors developed a concept similar to the one described previously but the technique utilized for production of the carrier signal and the phase demodulation technique are different. To generate the carrier signal, the continuous light from the laser converted into pulse light through an OM is split into two via an optical coupler to obtain a pulse-pair. An electro-optic phase modulator placed in one of the branches modulates one of the pulse pair and another optical coupler recombines the two pulse signals before being sent into the FUT. The interference signal from the two pulses and detected by the PD is effectively sinusoidally phase modulated. Experimental results demonstrate a positive influence caused by the selective phase modulation of the pulse-pair on the Rayleigh back-scattered signal. The demodulation algorithm utilized here is the PGC-DMS [60]. Unlike the PGC-Atan, the PGC-DMS demodulation technique does not need to apply the phase unwrapping, which algorithm is computationally intensive especially in interferometry system and by the way avoids also the eventual demodulation errors due to wrong unwrapping caused by the fluctuation of the carrier amplitude.

In contrast to the most popular Φ-OTDR vibration sensor system based on the differential intensity with a nonlinear response to external vibration, Fan et al. suggested a statistical analysis of the phase signal extracted from a Φ-OTDR system to locate the external vibration having a linear response to the magnitude of the applied vibration [61,62,63]. In coherent detection Φ-OTDR, the phase extracted signal *φ_ext_*(*t*) can be expressed as:*φ_ext_*(*t*) = *θ*(*t*) + *φ*(*t*) − *φ*_0_(14)
where *θ*(*t*) represents the vibration applied to the fiber under test and *φ*(*t*) − *φ*_0_ is the laser phase noise. The principle of the method is to determine the variance of a certain number of phase demodulated signals. In the absence of external action, the detected variance is nothing else than the variance of the laser phase noise that is a slowly varying quantity. By contrast, an external vibration causes an abrupt change in the variance of the phase signal such that the step change is proportional to the magnitude of the applied vibration. The experimental results confirm the consistency of the method and the linear response of the system. There are two important noise sources to the system: the detection white noise and the laser phase noise which becomes more serious with the fiber delay between the probe and reference lights in the coherent detection so, it appears to be necessary to compensate the phase noise in the system before extending the sensing range [62,63]. Fixing some FC/PC connectors as weak reflection points at certain positions of the FUT and defining them as the phase signal reference effectively suppresses the laser phase noise and the sensing range has been extended to 31 km [62]. The calculation of the variance of the collected phase difference signal is time consuming and the assistance with FPGA for signal processing will make the system sensing performance quite better.

## 8. Conclusions

In this paper, we have summarized the recent progress in the performance evaluation factors of Φ-OTDR sensor systems. We addressed this issue by a brief introduction of the conventional Φ-OTDR system, then we reported some methods and techniques to improve the *SNR* in Φ-OTDR systems such as the coherent detection, the moving averaging and moving differential, the laser frequency drift compensation, the enhancement of the *ER* of the pulse probe light, the pulse coding and several other signal processing methods. The techniques to increase the power of the Rayleigh back-scattered signal and by the way enable to enlarge the sensing range have been developed. These techniques include the Raman and the Brillouin amplification. Some new features in Φ-OTDR-related systems such as the OPCR and CPA Φ-OTDR to improve the *SR* in the range of the sub-meter, and the FDM Φ-OTDR to extend the frequency response bandwidth in both coherent-detection and direct-detection systems over the limit imposed by the pulse repetition rate have been discussed too.

The last item discussed in the topic was the capability of a Φ-OTDR system to perform quantitative measurements. Different methods involving the digital coherent detection, the I/Q demodulation, the phase diversity technique, the PGC demodulation, and the linear chirped-pulse reflectometry have been proposed for the purpose to rebuild the external vibration applied the FUT. Many issues for the compensation of the refractive index change caused by external action by an optical frequency shift of the probe light in order to measure precisely temperature/strain have been reviewed. Finally, a Φ-OTDR system using the phase extraction method having a linear response to external vibration has also been reported.

## Figures and Tables

**Figure 1 sensors-19-01709-f001:**
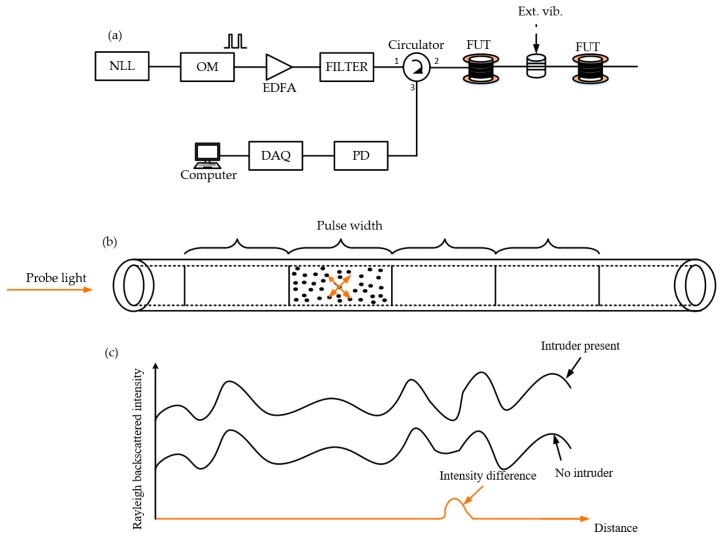
(**a**) Setup of a Φ-OTDR system, (**b**) illustration of Rayleigh phenomenon, (**c**) intrusion detection. NLL: narrow line-width laser, OM: optical modulator, EDFA: erbium-doped fiber amplifier, DAQ: data acquisition card, PD: photo-detector.

**Figure 2 sensors-19-01709-f002:**
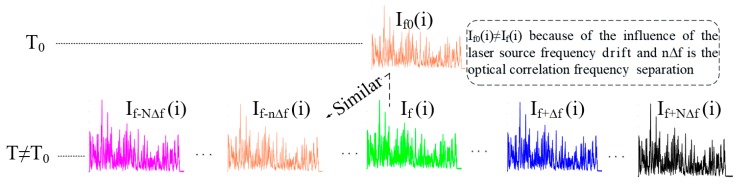
Principle of the active compensation of the laser output frequency drift using the wavelength diversity technique. *N* and *Δf* are respectively the operating sweep number and frequency sweep step of the tunable laser, n is the number corresponding to the OTDR trace best correlated to the previous.

**Figure 3 sensors-19-01709-f003:**
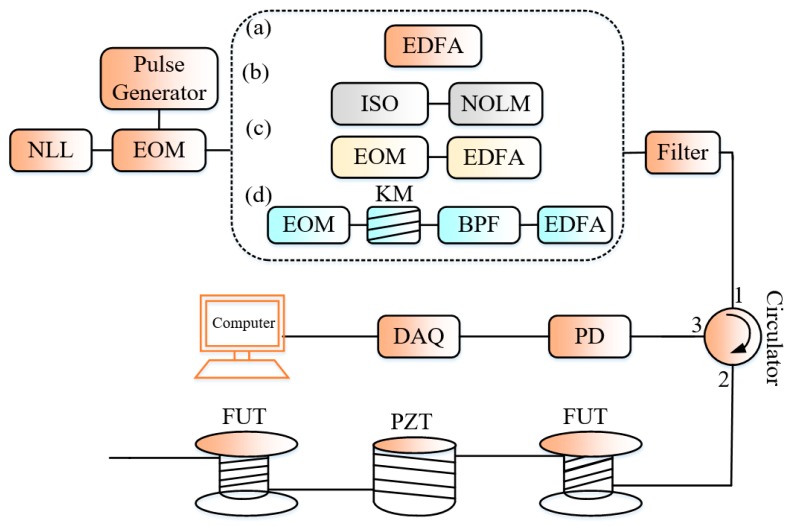
Experimental setup for comparison of the Φ-OTDR system basing on the *ER*: (**a**) conventional scheme; and higher *ER* schemes with: (**b**) NOLM: non-linear optical loop mirror (**c**) two-cascaded EOMs and (**d**) the nonlinear Kerr effect respectively. ISO: optical isolator, KM: Kerr medium, the other acronyms are defined previously.

**Figure 4 sensors-19-01709-f004:**
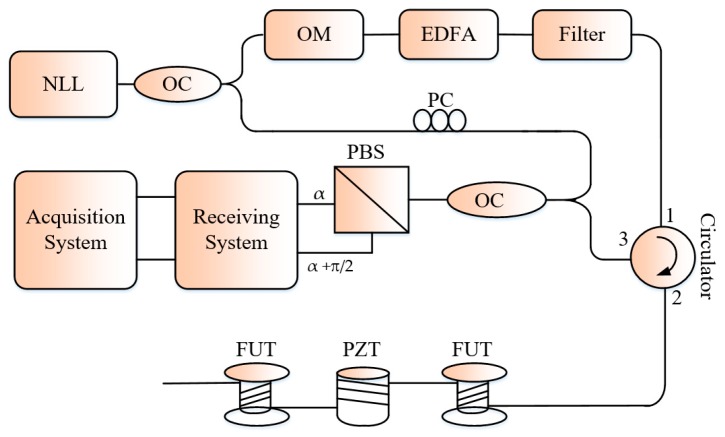
Schematic of the polarization diversity technique in Φ-OTDR system. OC: optical coupler PC: polarization controller.

**Figure 5 sensors-19-01709-f005:**
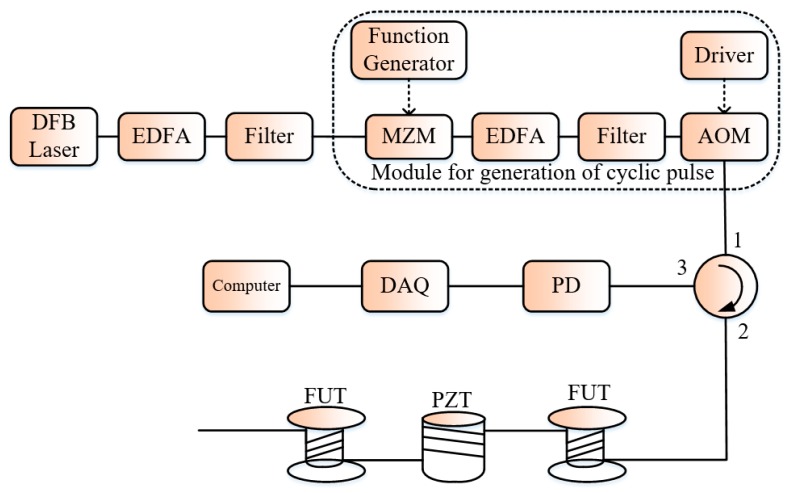
Experimental setup of the proposed pulse coding in Φ-OTDR system. MZM: Mach-Zehnder modulator.

**Figure 6 sensors-19-01709-f006:**
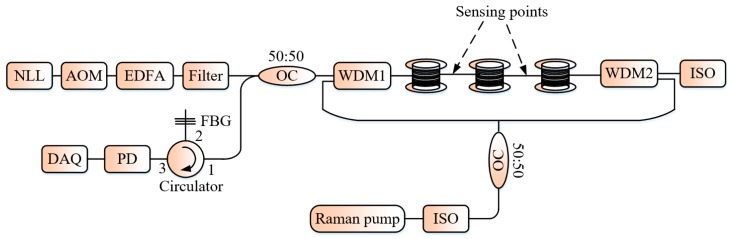
Experimental system of a Φ-OTDR system assisted with Raman amplification. WDM: wavelength division multiplexing, FBG: fiber Bragg grating filter.

**Figure 7 sensors-19-01709-f007:**
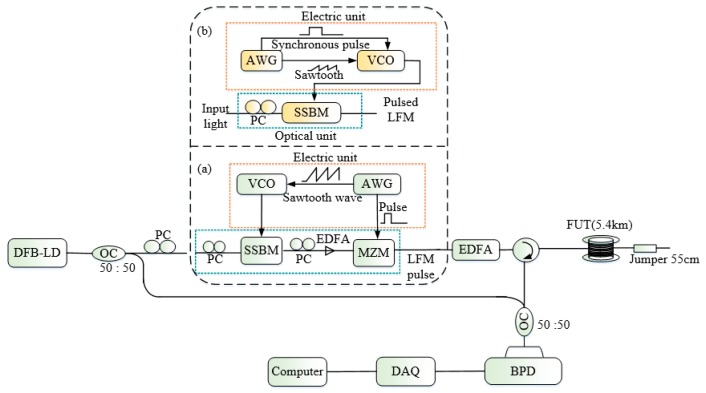
Experimental setup of the pulse-compression Φ-OTDR with (**a**) the primitive scheme for generation of the LFM pulse, (**b**) the improved scheme. SSBM: single sideband modulator, VCO: voltage-controlled oscillator, AWG: arbitrary waveform generator.

**Figure 8 sensors-19-01709-f008:**
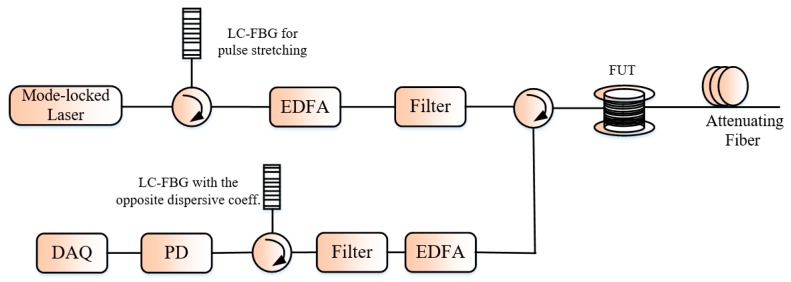
Setup of the chirped pulse Φ-OTDR system. LC-FBG: linearly chirped fiber Bragg grating.

**Figure 9 sensors-19-01709-f009:**
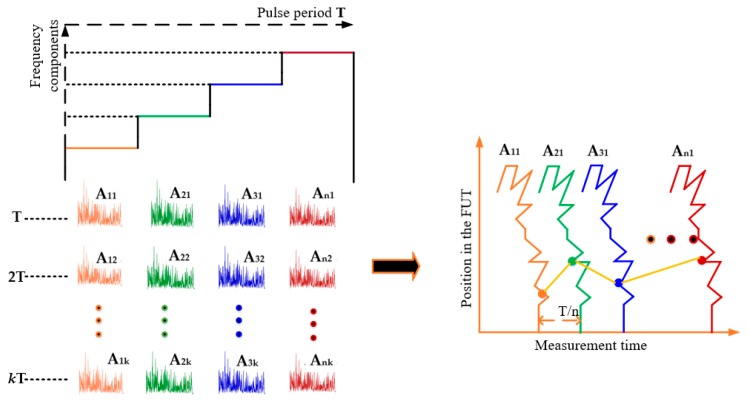
Signal processing for vibration measurement in FDM Φ-OTDR system.

**Figure 10 sensors-19-01709-f010:**
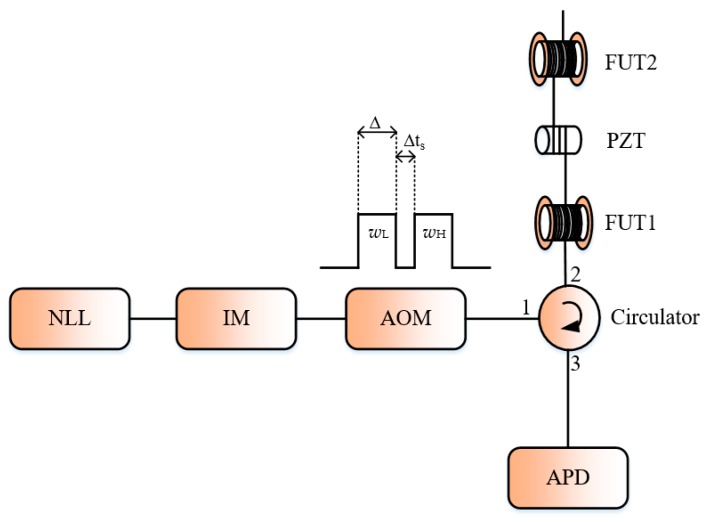
Experimental setup for the implementation of the FDM in direct-detection Φ-OTDR system. *w*_L_ and *w*_H_ are the frequency shifts introduced by the AOM in the dual-pulse pairs. Δ is the pulse-width, Δt_s_ is the double pulse spacing time. IM is an intensity modulator and APD is an avalanche photo-detector.

**Figure 11 sensors-19-01709-f011:**
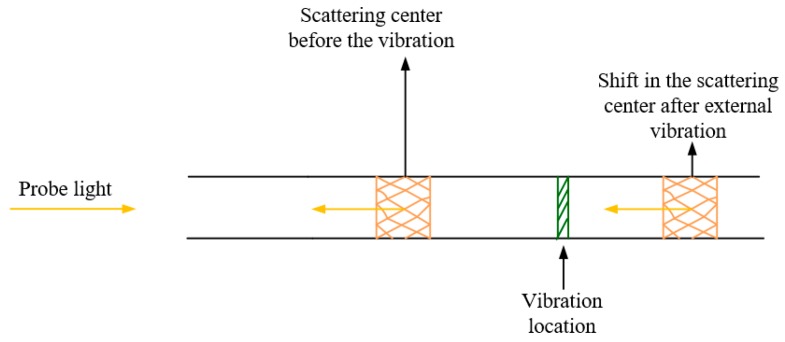
A concept of dynamic strain detection in Φ-OTDR based sensor.

**Figure 12 sensors-19-01709-f012:**
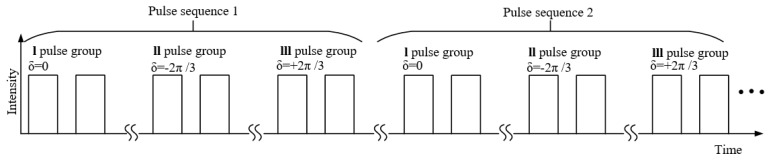
Pulse pair sequences with a phase shift of 0, +2π/3, −2π/3 respectively in the second pulse.

**Figure 13 sensors-19-01709-f013:**
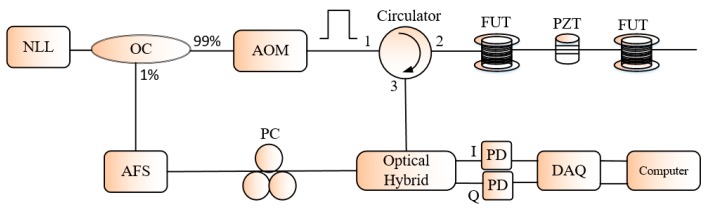
Setup of the Φ-OTDR based sensor using the 90° optical hybrid for phase demodulation. AFS: acoustic frequency shifter.

**Figure 14 sensors-19-01709-f014:**
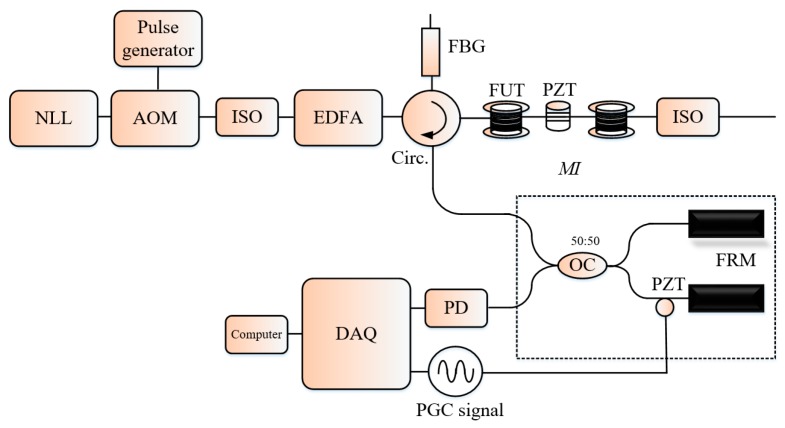
Setup of the Φ-OTDR with a MI at the receiving for generation of the PGC signal. FRM: Faraday rotation mirror.

**Table 1 sensors-19-01709-t001:** Working conditions of the pump schemes: (**a**) for the Brillouin amplification, (**b**) for the Raman amplification.

(**a**)	**Wavelength (nm)**	α **(dB/km)**	$g_{B}$ **(** $W^{-1}km^{-1}$ **)**	ε **(** $km^{-1}$ **)**
	1550	0.20	0.13 × $10^{3}$	4.3 × $10^{-5}$
(**b**)	**Wavelength (nm)**	α **(dB/km)**	$g_{R}$ **(** $W^{-1}km^{-1}$ **)**	ε **(** $km^{-1}$ **)**
	1455	0.27	0.35	$6\times 10^{-5}$
	1550	0.20	-	4.3 × $10^{-5}$

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
