# Peer review of "Recent Progress in the Performance Enhancement of Phase-Sensitive OTDR Vibration Sensing Systems"

_sensors, 2019, doi:10.3390/s19071709_

Reviewer 1 Report

The manuscript needs a moderate revision before consider it for publishing. 

Recently, a few new methods based on continuous-wave light sources were presented as phase-sensitive reflectometers operating in frequency and correlation domain to detect vibrations. The title of the manuscript should be corrected mentioning that the paper reviews only "impulse"-based systems, or, which is better, to include in introduction a brief description on CW-based systems, see:  a) DOI:10.1364/OE.22.008823     b)DOI:10.1364/OE.23.030347

I would recommend to include in p.2 "Basic operating principles.."  a brief introduction in  origins of noise in such a systems, since it is the most important limiting factor.  Discussion on averaging that reduces frequency range is  trivial....  

Please revise the text. Sometimes it is difficult to understand what authors want to say. See line 185, for example: ...a noise arising from the behavior of the light propagating along FUT,    and so on.

Author Response

Thanks very much for your valuable comments concerning our manuscript. These comments are very helpful for revising and improving our paper. Close attention have been paid to and a point per point correction has been provided accordingly. Our responses to your comments can be viewed in the document in attachment. Thanks a lot!

Reviewer 2 Report

This paper reviews about phase-sensitive OTDR for Distributed measurement of Acoustic vibration Sensing (DAS).

It is good review about the papers for them.

However, the authors have to correct and revise the paper.

The title of reference [8] is incorrect.

The reference [9] and [10] are not referred in the body of the paper.

The reference [19] and [20] are not explained in the body of the paper.

There is an error description at reference [22] in the "Reference" at the end of the paper.

The references [54] to [56] are only introduced in summary and the differences in them are not explained.

The author of [60] is incorrect.  I don't know whether the referenced paper or authors of the paper is incorrect to the really correct description. 

The reference [62] are not explained in the body of the paper.

There are many revised point in the description of references. 

I think there are probably more errors in the description of references in both body and "References" in the end of the paper.

The authors must check the all description in the paper and must revise the paper for unmistakable description because the reference description is most important thing in the review paper and the authors must respect the all authors and papers refferred in the paper.

Author Response

Thank you very much for your valuable comments concerning our manuscript. These comments are very helpful for revising and improving our paper. Careful attention has been paid to your comments and a point to point corrections have been made accordingly. Our responses to the comments can be seen in the document in attachment. We hope the revised version of our manuscript will meet your approval. Thanks a lot!

Reviewer 3 Report

This paper presents an overview of the recent progress in terms of performance enhancement in the phase-sensitive OTDR vibration sensing system. The paper is well-organized and provides comprehensive review of the topic for readers in related field. The paper can be accepted for publication through minor revisions.

Give the full name of OTDR in its first appearance, both in Abstract and main text. Check for any mistakes in abbreviations.

Provide clear version of Figure 2 if possible.

Differentiate 'PC' for 'personal computer' and 'PC' for 'polarization controller'. For example, in Figure 7 and Figure 13.

Author Response

Thanks very much for your valuable comments concerning our manuscript. These comments are very helpful for revising and improving our paper. Close attention have been paid to and a point per point correction has been provided accordingly. Our responses to your comments can be viewed in the document in attachment. Thanks a lot!

Round  2

Reviewer 1 Report

The revised manuscript can be considered for publishing in Sensors MDPI

Reviewer 2 Report

The paper was corrected very properly.  The authors revised the paper honestly.

I think the paper should be accepted and published in "Sensors".